# The influence of hydrogel stiffness on axonal regeneration after spinal cord injury

Haiyang Deng[1], Jiaqing Zhou[1], Cong Long[1], Ping Luo[1], Zhong Xiang[1], Hao Zeng [2]*

1 Department of Spinal Surgery, The Fourth Hospital of Changsha (Integrated Traditional Chinese and Western Medicine Hospital of Changsha, Changsha Hospital of Hunan Normal University), Changsha, Hunan, China, 2 Department of Spine Surgery Zone 2, The Affiliated Changsha Central Hospital, Hengyang Medical School, University of South China, Changsha, China

* 2621132371@qq.com

## Abstract

The core challenge in spinal cord injury(SCI) treatment is promoting axonal regeneration and forming new neural connections in damaged areas. However, mature CNS neurons have limited regenerative capacity, causing long-term dysfunction. Axonal regeneration involves elongating axons guided by growth cones, which sense and respond to external mechanical signals, integrating them into cytoskeleton reconstruction. After injury, growth cones experience altered mechanical forces due to changes in ECM stiffness. However, systematic studies on matrix stiffness's impact on axonal regeneration post-SCI remain insufficient. This study investigates the influence of hydrogel stiffness on axonal regeneration following SCI. Using gelatin methacryloyl (GelMA) hydrogels with varying stiffness levels, we cultured dorsal root ganglia (DRG) neurons in vitro and applied the hydrogels to a complete transection SCI mouse model. Results demonstrated that higher stiffness GelMA (15% w/v) significantly enhanced axonal extension and sensory functional recovery compared to lower stiffness (7.5% w/v). The study highlights the critical role of ECM stiffness in regulating axonal regeneration and suggests that optimizing hydrogel stiffness can promote neural regeneration and functional recovery after SCI. These findings provide valuable insights for developing therapeutic strategies in SCI treatment.

## Introduction

Spinal cord injury (SCI) is a permanent condition caused by direct tissue trauma, resulting in the loss of motor, sensory, and autonomic functions [1]. As a major global health concern, SCI contributes to over 700,000 new cases annually [2]. Current treatment strategies primarily focus on patient stabilization, complication prevention, and physical rehabilitation [3]. Despite advancements in clinical management, the recovery outcomes for SCI remain limited, underscoring the urgent need to explore alternative therapeutic approaches.

**Data availability statement:** All relevant data are within the manuscript and its Supporting information files.

**Funding:** This study was supported by the National Natural Science Foundation of China (grant no. 82160420) and Open Project of Guangxi Key Laboratory of Regenerative Medicine, Guangxi Medical University(grant no.202203).

**Competing interests:** The authors have declared that no competing interests exist.

The pathological process of SCI can be divided into two main phases: primary injury and secondary injury. Primary injury results from external forces causing spinal misalignment or direct destruction of spinal tissue, leading to morphological and structural damage to neurons and oligodendrocytes, as well as disruption of the blood-spinal cord barrier, which triggers a cascade of subsequent reactions [4]. Secondary injury involves complex biochemical and cellular changes, including alterations in cell permeability, activation of apoptotic pathways, and hypoxia, which further exacerbate spinal cord damage during the acute phase following injury [5]. Additionally, the accumulation of harmful chemical substances such as ischemia, intracellular electrolyte imbalances, reactive oxygen species, and peroxynitrite release contributes to cellular damage [6,7]. Vascular injury-induced inflammatory responses can lead to spinal cord edema, expanding the area of damage. Over time, the acute inflammatory response gradually subsides, and the spinal cord enters a repair phase, which includes vascular remodeling, remyelination of nerve fibers, and neural network reconstruction [8–10]. However, activated glial cells and macrophages secrete inhibitory proteins that form glial scars, which, while limiting the injury area, also impede neural regeneration [11].

The core challenge in SCI treatment lies in promoting axonal regeneration and the formation of new neural connections in the damaged area. However, the regenerative capacity of neurons in the mature central nervous system is significantly reduced, leading to long-term neurological dysfunction [12]. Axonal regeneration is a complex process, where developing axons elongate and sense their surroundings to achieve target-oriented growth. The growth cone, as the leading structure of the axon, can sense and respond to external mechanical signals from the environment and integrate these signals into the reconstruction of the internal cytoskeleton to regulate axonal regeneration [13,14]. In the case of neuronal injury, the mechanical properties of the regenerating growth cone change due to alterations in the stiffness of the extracellular matrix. During axonal regeneration, growth cones can sense and respond to mechanical signals in the environment, while the formation of glial scars may alter the mechanical properties of the local microenvironment [15–17]. Natural polymer-based hydrogels, particularly gelatin methacryloyl (GelMA), have shown great potential in tissue engineering due to their excellent biocompatibility, tunable rheological properties, and photocrosslinking capabilities [18–20]. Studies have demonstrated that dorsal root ganglion (DRG) neurons exhibit enhanced regenerative capacity on stiffer substrates, suggesting that extracellular matrix (ECM) stiffness plays a critical role in regulating axonal regeneration [21]. However, systematic research on the effects of substrate stiffness on axonal regeneration following SCI remains insufficient. Based on this, we hypothesize that the stiffness of GelMA can modulate neuronal behavior to promote axonal regeneration, thereby improving neurological function after SCI.

In this study, we first cultured DRG on GelMA of varying stiffness in vitro, followed by the application of these hydrogels to a SCI mouse transection model. The results demonstrated that GelMA with higher stiffness significantly enhanced axonal extension and sensory functional recovery in SCI mice. These findings reveal the

coordinated cellular responses during regeneration and provide new insights into developing therapeutic strategies to promote neural regeneration after SCI.

## Methods and materials

### Preparation and characterization of GelMA

GelMA precursor was purchased from Engineering for Life Company (EFL-GM-30). A specified mass of GelMA was weighed and dissolved in deionized water at 50°C to prepare GelMA solutions of varying concentrations. The GelMA solution was then mixed with the photoinitiator lithium phenyl-2,4,6-trimethylbenzoylphosphinate (LAP) to achieve a final concentration of 0.025% w/v. The mixture was heated in a 37°C water bath and sterilized using a 0.22 μm filter. Subsequently, the solution was crosslinked by exposure to ultraviolet (UV) light. The structural and morphological changes of the hydrogels were examined using a Mira3-TESCAN scanning electron microscope (SEM).

### Nanoindentation

The hydrogel precursor solution (250 μL) was first confined in a cylindrical mold and exposed to ultraviolet (UV) light from all sides for 60 seconds. Following the polymerization step, the samples were firmly attached to glass slides using a high-strength adhesive. Nanoindentation experiments were conducted in 1 × phosphate-buffered saline (PBS) using a Chiaro nanoindenter (Optics11, Netherlands), equipped with a cantilever of 0.45 N/m stiffness and a spherical tip with a radius of 49.5 μm. The effective indentation modulus (E) at each indentation point was determined by analyzing the slope of the loading segment of the force-depth curve, in accordance with established protocols.

### Isolation and culture of DRGs

Dorsal root ganglia (DRGs) were harvested from 8-week-old mice. For euthanasia, an overdose of sodium pentobarbital (150 mg/kg) was administered. The spinal column was isolated and bisected along the sagittal plane, followed by the removal of the spinal cord. DRGs were carefully dissected from the intervertebral foramina, and the nerve roots were trimmed, leaving the DRG bodies intact. The DRGs were then bisected and cultured in neurobasal medium (Gibco, USA) supplemented with 2% B-27 (Gibco, USA), 1% penicillin/streptomycin (Gibco, USA), and nerve growth factor (50 ng/ml, Invtrogen). The plates were placed in an incubator maintained at 37°C and 5% $CO_2$ for 30 minutes to allow the DRGs to adhere to the culture surface. After adhesion, culture medium was added, and the DRGs were cultured for 72 hours. Following the culture period, the DRGs were analyzed using immunofluorescence staining, with a particular focus on the expression of neurofilament (NF).

### Complete transection SCI model

All experimental animal techniques were approved by the Ethics Committee of The Fourth Hospital of Changsha of Hunan Normal University for Scientific Research. A total of 60 Adult female C57BL/6 mice (8 weeks old) were randomly assigned to the following groups: Sham group (n = 12), SCI group (no hydrogel implantation, n = 15, 3 died postoperatively), 7.5% Hydrogel group (n = 12), 15% Hydrogel group (n = 12) and 9 mice for DRGs culture. Anesthesia was induced by intraperitoneal injection of 0.3% sodium pentobarbital (60 mg/kg). A midline incision was made on the dorsal side, and the spinal cord was carefully exposed following a T8 laminectomy. A 1 mm segment of the spinal cord was completely transected and removed. Subsequently, 10 μL of hydrogel at different concentrations was locally injected into the injury site, followed by ultraviolet (UV) light exposure to solidify the gel. The muscle and skin were sutured sequentially. To minimize pain and distress, local lidocaine injections were applied at the incision site before surgery. Postoperatively, ibuprofen (10 mg/kg) was administered daily for 7 days to alleviate inflammation and pain. Manual bladder emptying was performed twice daily until bladder function was restored.

## Basso Mouse Scale (BMS) Scoring

Two evaluators, blinded to the experimental group assignments, quantified the motor function of the mice using the Basso Mouse Scale (BMS) scoring system. Assessments were conducted before SCI, immediately after injury, and at subsequent time points (1d, 3d, 1w, 2w, 3w, 4w, 5w, 6w, 7w and 8w days post-injury). During the scoring process, mice were placed in an open field and allowed to move freely for 4 minutes to observe their motor functional recovery. The BMS score ranges from 0 to 9, where 0 indicates complete paralysis of the hindlimbs and 9 represents normal motor function. The scoring system comprehensively evaluates multiple aspects of hindlimb movement, including joint mobility, weight-bearing ability, plantar stepping, coordination, paw position, and trunk and tail movement control.

## Tissue perfusion and staining

For hematoxylin and eosin (H&E) staining, after perfusing the left ventricle with saline, the mice were subsequently fixed using a 4% paraformaldehyde solution. The spinal cord, heart, liver, spleen, lungs, kidneys, and bladder were then collected and dehydrated for sectioning. For H&E staining, the spinal cord was embedded in paraffin, and sagittal sections with a thickness of 5 micrometers were prepared and stained accordingly. The injured areas of the spinal cord and the thickness of the bladder muscle tissue were manually outlined and quantified using ImageJ software.

The immunofluorescence staining steps are as follows: Place the specimen to be stained into the blocking and permeabilizing solution (1% BSA and 0.3% Triton X-100 in PBS buffer) and incubate at room temperature for 2 hours. After that, remove the specimen, dry the residual liquid with filter paper, and place it into a humidified chamber. Then, cover the sample completely with the diluted primary antibody (NF antibody: 1:400; GFAP antibody: 1:400) and incubate in the dark at 4°C overnight. On the second day, take out the humidified chamber from the 4°C refrigerator and allow it to warm up at room temperature for 20 minutes. Soak the sections in 0.3% Triton X-100 (PBS buffer) for 3 washes, each for 5 minutes, to remove the residual primary antibody. Dilute the corresponding fluorescent secondary antibody at 1:400 and cover the specimen completely. Incubate in the humidified chamber at room temperature in the dark for 1.5 hours. Soak the specimen in 0.3% Triton X-100 (PBS buffer) for 3 washes, each for 5 minutes, to remove the residual secondary antibody. Finally, take out the specimen, dry the residual liquid with filter paper, add DAPI working solution (containing antifade mounting medium). The specific information of the fluorescent antibodies is shown in Table S1. The images were captured using a fluorescence microscope (Carl Zeiss, Germany).

## Statistical analysis

In this study, experimental data were statistically analyzed and processed using Prism 9 software. The results are presented as the arithmetic mean of the samples plus or minus the standard deviation (SD). For comparisons between two independent sample groups, an unpaired t-test was used. For comparisons involving three or more groups, one-way analysis of variance (ANOVA) was performed, followed by post-hoc analysis using Tukey's multiple comparison test. For datasets involving repeated measurements, such as BMS scores and NF or GFAP expression levels, two-way ANOVA was employed as the statistical method. In all statistical tests conducted in this study, a P-value of $<0.05$ was considered the threshold for statistical significance.

## Results

### Material characterization of hydrogels

To investigate the influence of stiffness on neuronal behavior and function, we fabricated GelMA with gradient elastic moduli by varying the GelMA concentration. Cells are sensitive to the stiffness of the underlying substrate and surrounding microenvironment, which significantly influences their behavior. Therefore, hydrogels with tunable stiffness have garnered considerable attention in biomedical applications. Initially, we examined the microstructure of the hydrogel materials using

scanning electron microscopy (SEM). The results revealed that hydrogels of different concentrations exhibited a uniform porous structure, consistent pore wall homogeneity, smooth surface texture, and excellent pore interconnectivity (Fig 1A). Subsequently, we measured their stiffness using nanoindentation. The results demonstrated that increasing the GelMA concentration from 7.5% to 15% w/v led to a significant increase in Young's modulus, with values of $15.12 \pm 0.46$ kPa and $51.45 \pm 1.11$ kPa, respectively (Fig 1B). Thus, hydrogels with GelMA concentrations of 7.5% and 15% w/v were classified as soft and stiff substrates, respectively. These findings indicate that the stiffness of GelMA can be effectively controlled by adjusting the GelMA concentration. To mimic the typical placement of cells on scaffold surfaces in tissue engineering, we employed a method of seeding DRG onto the surface of GelMA hydrogels (Fig 1C). This approach allows for a controlled environment to study cell behavior and interactions with the hydrogel matrix, providing valuable insights into the potential applications of GelMA hydrogels in regenerative medicine and tissue engineering strategies.

### In vivo characterization of hydrogels

To evaluate the biocompatibility and potential toxicity of the GelMA, GelMA of varying concentrations were implanted into the injury sites of mice with SCI. The impact on major organs was assessed through histological analysis. H&E staining was employed to examine tissue sections of the heart, liver, spleen, lungs, and kidneys (Fig 2A). The cardiac muscle fibers were regularly arranged with centrally located nuclei and distinct intercalated discs, and no inflammatory cell infiltration was observed in the interstitium. The hepatic lobules maintained intact structures, with hepatocytes presenting as polygonal shapes and abundant cytoplasm, and the central veins and hepatic sinusoids were clearly visible. The spleen exhibited intact capsule and trabeculae, with white pulp composed of lymphocytes and red pulp containing a large number of red blood cells, and the marginal zone was distinct. The lungs showed thin alveolar walls and alveolar cavities devoid of exudates, with clear structures of bronchi and bronchioles and intact mucosal epithelium. The kidneys displayed

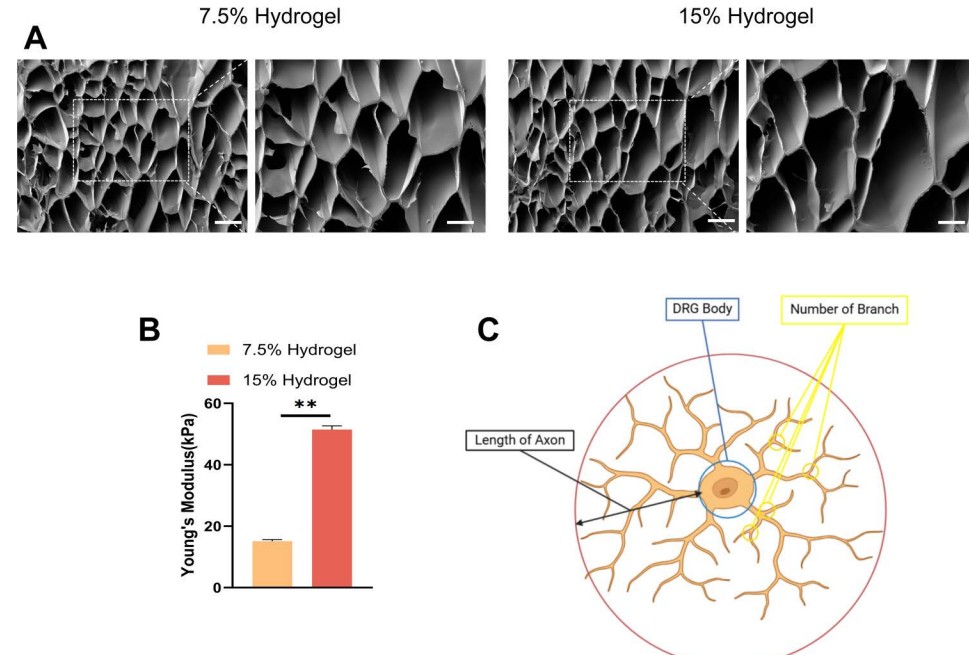

**Fig 1. Material characterization of hydrogels.** A. Mira3-TESCAN SEM images showed the microstructures of the hydrogel. Scale bar: 100 µm. Magnification: 100X. Scale bar, 50 µm (enlarged view). B. The Young's modulus (stiffness) of GelMA hydrogel with different concentrations, **p < 0.01. C. Quantitative schematic of dorsal root ganglion (DRG) axon length and number of branches.

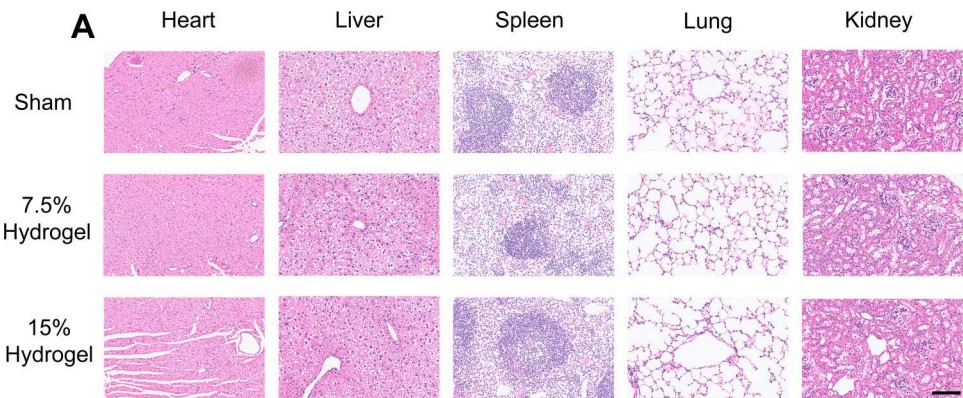

**Fig 2. In vivo characterization of hydrogels.** A. Representative images of HE- staining of the heart, liver, spleen, lung, and kidney tissue showed no differences between the Sham, 7.5% Hydrogel and 15% hydrogel treated mice groups. Scale bar, 100μm. Magnification: 40X.

normal glomerular structures, with neatly arranged renal tubules and collecting ducts, and no inflammatory cell infiltration was detected in the interstitium. The experimental results revealed no significant pathological alterations in the tissue architecture of these organs following the implantation of GelMA at different concentrations.This indicates that the GelMA exhibited excellent biocompatibility and did not induce adverse effects on vital organs, supporting their potential for safe application in SCI treatment.

### In vitro evaluation of axonal growth promotion by GelMA of different concentrations in DRG

DRG were cultured as an in vitro model to study axonal growth, and axons were labeled with anti-neurofilament (NF) antibody for immunofluorescence analysis (Fig 3A). The maximum extension distance of axons and the number of dendritic branches were quantified using fluorescence microscopy. The results revealed that the 15% GelMA group exhibited significantly longer maximum axonal extension distances ($1166.00 \pm 46.09$ vs $725.17 \pm 28.69$) and a greater number of axonal branches ($230.00 \pm 17.23$ vs $145.17 \pm 11.19$) compared to the 7.5% GelMA group (Fig 3B). These findings indicate that GelMA with higher stiffness possess a stronger ability to enhance axonal regeneration.

### Motor function assessment treated with hydrogels of different concentrations after SCI

In the vivo experiments, we established a spinal cord injury fully transection SCI model (Fig 4A). The Basso Mouse Scale (BMS) was employed to assess motor function recovery at multiple time points before surgery and after injury (1d, 3d, 1w, 2w, 3w, 4w, 5w, 6w, 7w, and 8w). The results indicated that all groups exhibited motor dysfunction following spinal cord injury. However, the 7.5% Hydrogel group demonstrated improved recovery compared to the Control group at 4w ($0.25 \pm 0.5$ vs $0.67 \pm 0.6$), 5w ($0.33 \pm 0.3$ vs $1.17 \pm 0.3$), 6w ($0.50 \pm 0.4$ vs $1.50 \pm 0.4$), 7w ($0.58 \pm 0.4$ vs $1.83 \pm 0.3$), 8w ($0.58 \pm 0.4$ vs $2.25 \pm 0.5$). Furthermore, starting from the 4th week post-injury, the 15% Hydrogel group showed superior motor function recovery compared to the 7.5% Hydrogel group at 2w ($0.42 \pm 0.5$ vs $0.83 \pm 0.4$), 3w ($0.58 \pm 0.5$ vs $1.25 \pm 0.5$), 4w ($0.67 \pm 0.6$ vs $1.83 \pm 0.5$), 5w ($1.17 \pm 0.3$ vs $2.17 \pm 0.4$), 6w ($1.50 \pm 0.4$ vs $2.33 \pm 0.4$), 7w ($1.83 \pm 0.3$ vs $3.00 \pm 0.5$), 8w ($2.25 \pm 0.5$ vs $3.42 \pm 0.4$) (Fig 4B).

### In vivo evaluation of axonal regeneration promotion by hydrogels of different concentrations after SCI

To further elucidate the role of GelMA stiffness in axonal regeneration followingSCI, this study employed immunofluorescence analysis for detailed observation and assessment. At 8w post-treatment in SCI mice, significant therapeutic

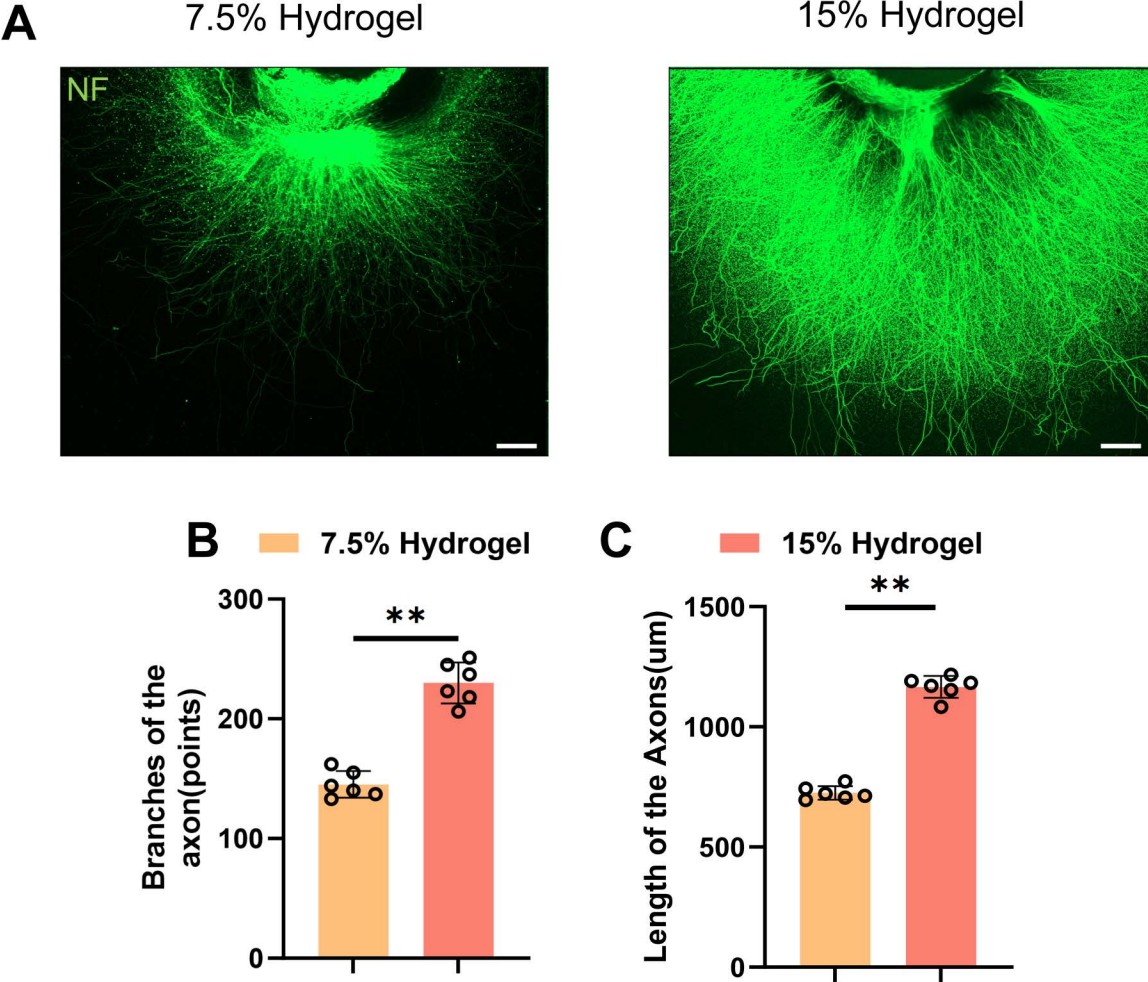

**Fig 3. In vitro evaluation of axonal growth promotion by GelMA of different concentrations in DRG.** A. Representative images of length of the axons and branches of the axons treated with 7.5% Hydrogel and 15% Hydrogel in DRG. Magnification: 40X. Scale bar 200 μm. B. Quantification of branches of the axons are shown in (A), n=6. C. Quantification results for length of the axons are shown in (A), n=6. The data was shown as a mean±SD, * P<0.05, ** P<0.01. (One-way ANOVA between multiple groups plus Tukey's hoc test).

effects were observed, particularly in the 7.5% GelMA and 15% GelMA groups. Immunofluorescence analysis revealed a notable increase in NF-positive signals within the injury site, with the 15% GelMA group showing the most pronounced enhancement (Fig 5A, B). The increased NF-positive signals indicate active axonal regeneration and suggest that the repair of damaged nerve fibers was promoted.These findings highlight the critical role of GelMA stiffness in facilitating axonal regeneration and provide evidence that stiffer GelMA, such as the 15% GelMA formulation, offer superior therapeutic potential for SCI repair.Concurrently, immunofluorescence co-localization analysis corroborated these findings (Fig 5C).

### Histological evaluation treated with hydrogels of different concentrations after SCI

At the 56-day post-SCI time point, spinal cord tissue samples were collected and subjected to H&E staining. Quantitative analysis was performed on the integrity of the spinal cord tissue and the size of the cavitation in the injury area. The study found that, compared to the 7.5% Hydrogel group and the Control group, the 15% Hydrogel group exhibited the smallest

**A**

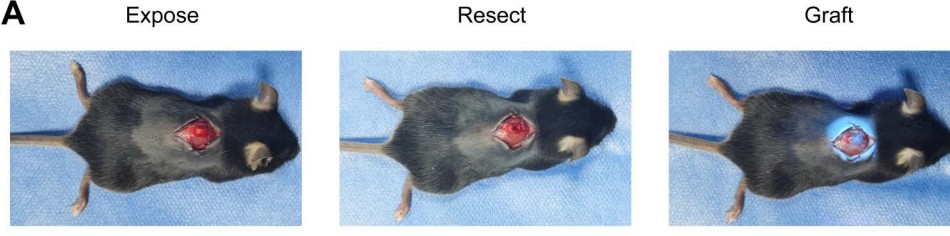

Expose    Resect    Graft

**B**

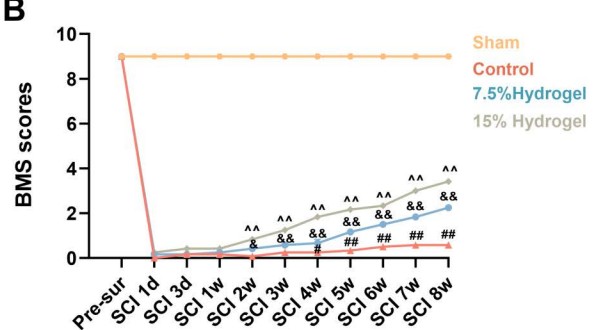

**Fig 4. Motor function assessment treated with hydrogels of different concentrations after SCI.** A. Schematic diagram of the modeling process for the complete transection model of SCI. **B.** BMS scores at different time points after SCI in the treatment groups. n = 6. The data was shown as a mean ± SD, & $P < 0.05$ 15% Hydrogel vs. 7.5% Hydrogel, # $P < 0.05$ 7.5% Hydrogel vs. Control, ^^ $P < 0.01$ 15% Hydrogel vs. Control, && $P < 0.01$, ## $P < 0.01$. (Two-way ANOVA plus Turkey'S hoc test between multiple groups).

cavitation area (12.34 ± 2.38 vs 29.34 ± 3.55 vs 60.82 ± 3.88, Fig 6A, C). This significant difference intuitively reflects the superior efficacy of the 15% hydrogel in promoting spinal cord tissue integration. It can efficiently connect the tissues on both sides of the spinal cord injury site, effectively reducing the formation of cavitations, thereby providing robust support for the repair and functional recovery following spinal cord injury.. Additionally, the thickness of the smooth muscle layer in bladder tissue was assessed via H&E staining (264.17 ± 33.16 vs 325.38 ± 14.72 vs 450.35 ± 16.84). These data clearly demonstrate that the 15% hydrogel group also performs remarkably well in alleviating neurogenic bladder injury. (Fig 6B, D). Overall, the 15% hydrogel group shows a significant advantage over the 7.5% hydrogel group in terms of functional recovery. This advantage is not only reflected at the histological level but also fully demonstrated at the functional level. These detailed descriptive information and precise numerical data provide a solid foundation for us to deeply understand the mechanism of action of the 15% hydrogel in spinal cord injury repair and also offer strong evidence for subsequent clinical application research.

## Discussion

After SCI, axonal regeneration of neurons faces significant challenges. Although axons possess an inherent regenerative capacity after injury, the recovery process is typically slow and limited, even with the use of nerve grafts to assist regeneration [22]. Studies have shown that the mechanical properties of the ECM, particularly matrix stiffness, play a critical role in regulating axonal regeneration [23]. However, the molecular mechanisms underlying stiffness-mediated regulation of axonal regeneration remain poorly understood.

All cells reside within a mechanical microenvironment, which includes the stiffness of the ECM or adjacent cells. The mechanical microenvironment plays a crucial role in regulating various physiological processes, including morphogenesis, cell migration, and tissue development [24–26]. The mechanical signals provided by the ECM are vital for neuronal

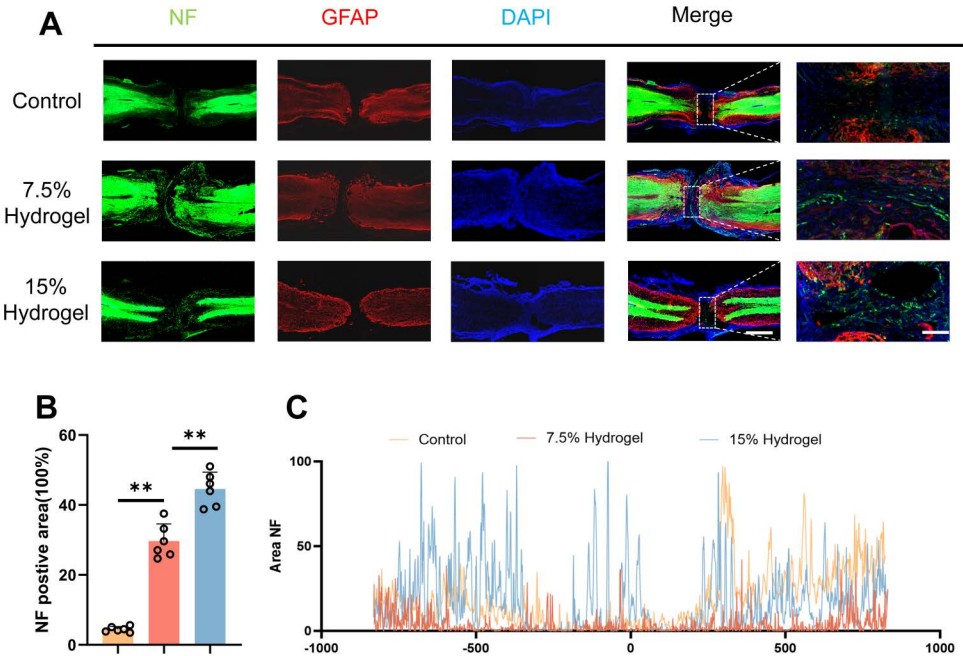

**Fig 5. In vivo evaluation of axonal regeneration promotion by hydrogels of different concentrations after SCI.** A. Representative images of NF (green-Alexa Fluor® 488) immunostained neurons and GFAP (red-Alexa Fluor® 594) astrocytes in mice treated with Control, 7.5% Hydrogel and 15% hydrogel after spinal cord injury. Magnification: 10X. Scale bar, 150 μm, Scale bar, 30 μm (enlarge view). B. Quantification of labeled neurons in different groups. Each group n = 6. C. Curves showing the continuous distribution of NF positive neuronal fiber area in (A).The data was shown as a mean ± SD, * P < 0.05, ** P < 0.01. (One-way ANOVA between multiple groups plus Tukey's hoc test).

function, promoting axonal growth and maturation [27]. Therefore, precise modulation of matrix stiffness holds promise for enhancing neuronal axonal regeneration after SCI. GelMA, as a versatile hydrogel, has been widely used in tissue engineering scaffolds and bioinks [28,29]. However, the mechanisms by which GelMA hydrogel stiffness affects neuronal behavior have not yet been fully elucidated.

Previous studies have demonstrated that appropriate matrix stiffness can effectively induce axonal growth and dendritic formation in DRG neurons [21,30]. Consistent with prior research, our findings underscore the significance of matrix stiffness in regulating axonal regeneration in SCI. In this study, we prepared 7.5% and 15% GelMA hydrogels by modulating the concentration of GelMA to identify the optimal matrix stiffness for promoting axonal regeneration at the injury site after SCI. The 7.5% GelMA hydrogel (15.12 kPa) was chosen to mimic the physiological stiffness of DRG tissue under physiological conditions, while the hard substrate environment of the 15% GelMA hydrogel (51.45 kPa) has been reported to effectively promote axonal regeneration of neurons in vitro. Therefore, we hoped to gain a more comprehensive understanding of the effects of different stiffness matrices on axonal regeneration of central neurons after SCI by comparing GelMA hydrogels of different concentrations. The experimental results revealed that 15% w/v GelMA (51.45 kPa) significantly facilitated the formation and ingrowth of well-structured neuronal axons at the injury center of the spinal cord, outperforming the lower stiffness level of 15.12 kPa. Moreover, compared to mice in the control group or other treatment groups, mice in the 15% w/v GelMA group exhibited superior hindlimb motor function recovery as indicated by BMS scores [31–33]. However, we did not further assess the degree of limb spasticity in the mice. We are well aware that the degree of limb spasticity is also an important clinical issue after spinal cord injury, which has a significant impact on patients' rehabilitation and quality of life. In future in-depth studies, we plan to include the assessment of limb spasticity in our research scope and use more comprehensive and multidimensional assessment methods to more fully evaluate the recovery of neurological

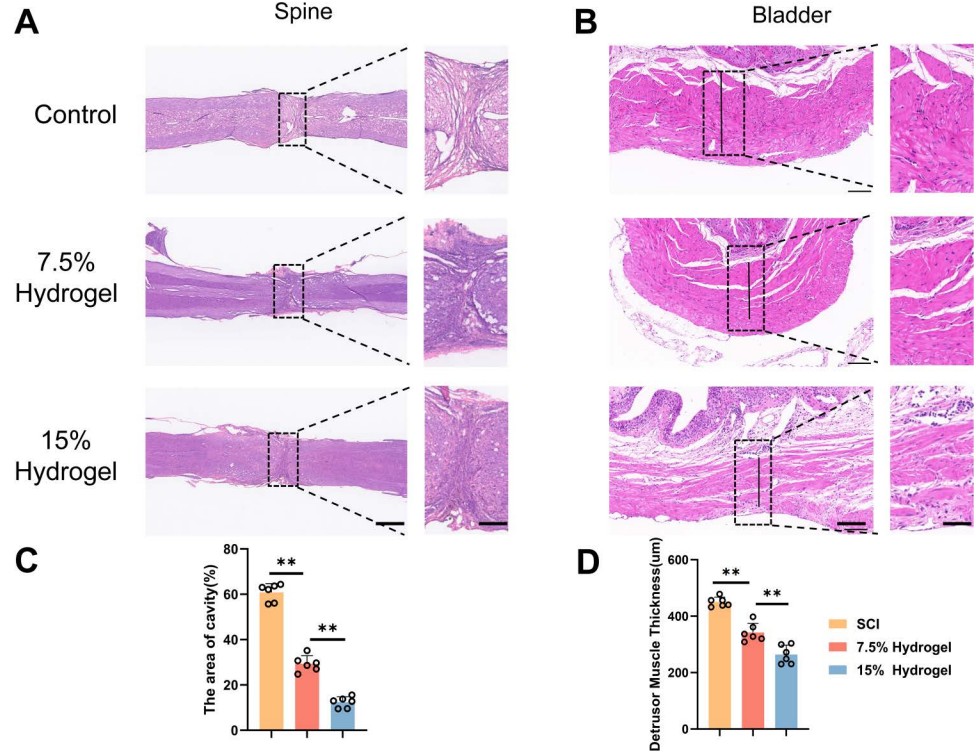

**Fig 6. Histological evaluation treated with hydrogels of different concentrations after SCI.** A. H&E staining of spine in each group at 56 days post-SCI. Scale bar, 150 μm. B. H&E staining of bladder sections from each group at 56 days post-SCI. The black line indicates the detrusor muscle. Magnification: 10X. Scale bar, 100 μm. C. Quantification results for cavity area are shown in (A), n = 6. D. Quantification of detrusor muscle thickness in (B), n = 6. The data was shown as a mean ± SD, * P < 0.05, ** P < 0.01. (One-way ANOVA between multiple groups plus Tukey's hoc test).

function after spinal cord injury. This finding supplements previous research on hydrogel-promoted neuronal ingrowth and, for the first time, clarifies the pivotal role of matrix stiffness in this process. Notably, our study also found that modulating matrix stiffness not only affects axonal regeneration but also has a significant impact on neuronal survival and functional recovery. Neurons cultured in 15% w/v GelMA hydrogel exhibited higher survival rates and stronger axonal extension capabilities, which is consistent with the observed improvement in neurological function recovery in in vivo experiments. This suggests that optimizing matrix stiffness can not only promote axonal regeneration but also enhance overall neuronal functional recovery. Furthermore, based on our findings, we propose that the loss of SCI axonal regeneration capacity may be attributed to the soft physiological stiffness of the spinal cord (0.1–1 kPa).

Our experiments have demonstrated that the stiffness of hydrogel matrices can influence the capacity for axonal regeneration of neurons at the injury site following SCI. However, the specific signaling pathways through which matrix stiffness exerts this effect remain to be elucidated. Existing research has shown that matrix stiffness can modulate axonal regeneration via multiple signaling pathways. For instance, Piezo1 can regulate axonal regeneration by sensing mechanical signals through the $Ca^{2+}$–CaMKII–FAK–actin cascade [14,21]. The PI3K/Akt/mTOR pathway promotes axonal regeneration by inhibiting GSK3β [34]. Integrins can transmit mechanical signals from the ECM via the FAK signaling pathway [35,36]. The interplay of these signaling pathways provides a multi-faceted regulatory mechanism for nerve regeneration and offers new insights for the treatment of SCI and other neurological disorders. Future work in our laboratory will focus on investigating the molecular mechanisms underlying the regulation of SCI neuronal regeneration by matrix stiffness, with an emphasis on these signaling pathways.

In conclusion, our study is the first to systematically reveal the critical role of GelMA stiffness in axonal regeneration after SCI. These findings provide important theoretical and experimental support for developing GelMA stiffness-based strategies for neural regeneration. Future research will further optimize the stiffness and other physicochemical properties of GelMAs to advance their clinical application in SCI repair.

## Supporting information

**S1 Table. Antibody catalog.**
(DOCX)

## Author contributions

**Conceptualization:** Haiyang Deng, Jiaqing Zhou, Hao Zeng.

**Data curation:** Haiyang Deng.

**Formal analysis:** Haiyang Deng.

**Funding acquisition:** Hao Zeng.

**Investigation:** Haiyang Deng, Ping Luo, Zhong Xiang.

**Methodology:** Haiyang Deng, Jiaqing Zhou, Cong Long, Zhong Xiang.

**Project administration:** Haiyang Deng.

**Resources:** Haiyang Deng.

**Supervision:** Cong Long, Ping Luo, Hao Zeng.

**Visualization:** Jiaqing Zhou, Cong Long.

**Writing – original draft:** Haiyang Deng, Ping Luo, Hao Zeng.

**Writing – review & editing:** Haiyang Deng, Zhong Xiang, Hao Zeng.

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
