## [Decision Letter · Decision Letter 0]

May 12 2025

Dear Dr. zeng,

Thank you for submitting your manuscript to PLOS ONE. After careful consideration, we feel that it has merit but does not fully meet PLOS ONE’s publication criteria as it currently stands. Therefore, we invite you to submit a revised version of the manuscript that addresses the points raised during the review process.

We look forward to receiving your revised manuscript.

Kind regards,

Pradeep Kumar

Academic Editor

PLOS ONE

Journal Requirements:

3. To comply with PLOS ONE submissions requirements, in your Methods section, please provide additional information regarding the experiments involving animals and ensure you have included details on (1) methods of sacrifice, (2) methods of anesthesia and/or analgesia, and (3) efforts to alleviate suffering.

This study was supported by the National Natural Science Foundation of China (grant no. 82160420) and Open Project of Guangxi Key Laboratory of Regenerative Medicine, Guangxi Medical University(grant no.202203).

This study was supported by the National Natural Science Foundation of China (grant no. 82160420) and Open Project of Guangxi Key Laboratory of Regenerative Medicine, Guangxi Medical University(grant no.202203)

This study was supported by the National Natural Science Foundation of China (grant no. 82160420) and Open Project of Guangxi Key Laboratory of Regenerative Medicine, Guangxi Medical University(grant no.202203)

6. We note that your Data Availability Statement is currently as follows: All relevant data are within the manuscript and its Supporting Information files.

7. We note you have included a table to which you do not refer in the text of your manuscript. Please ensure that you refer to Table S1 in your text; if accepted, production will need this reference to link the reader to the Table.

Reviewers' comments:

Reviewer's Responses to Questions

**Comments to the Author**

1. Is the manuscript technically sound, and do the data support the conclusions?

Reviewer #1: Yes

Reviewer #2: Partly

2. Has the statistical analysis been performed appropriately and rigorously?

Reviewer #1: Yes

Reviewer #2: I Don't Know

3. Have the authors made all data underlying the findings in their manuscript fully available?

Reviewer #1: No

Reviewer #2: Yes

4. Is the manuscript presented in an intelligible fashion and written in standard English?

Reviewer #1: Yes

Reviewer #2: Yes

Reviewer #1: Dear Authors,

The research topic is relevant and promising. The obtained results can be used for further comparative studies. The described studies were conducted using many modern methodological approaches, indicating the seriousness of the obtained results.

To improve the publication, please consider the following comments:

1) Section 2.3 p.5: Please indicate the culture medium composition for DRG neurons.

2) Section 2.4 p.5: Please indicate the percentage of animal survival after SCI modelling.

- How many animals were taken for each group?

- Add the average weight of the experimental animals to the text.

3) Section 2.5 p.6: Along with assessing motor function, it is imperative to assess the level of spasticity of the limbs, since these functional indicators are interrelated. Argue why there are no data on the assessment of spasticity in this experiment.

4) Section 2.6 p.6: Please indicate the characteristics of the secondary antibodies and add this information to the table on p.20.

- What solution(s) were used to wash the sections after primary and secondary antibodies? Indicate this information in the text.

- Was blocking solution (its components?) used before applying primary antibodies to nerve tissue sections?

5) Section 2.7 p.7: The number of analyzed samples is not indicated. Please add it to the text.

6) Section 3.2 p. 8, Fig.2: Describe in detail what means "The experimental results revealed no significant pathological alterations in the tissue architecture of these organs following the implantation…"

- How did this manifest itself in the indicated organs, what was the morphology of these organs?

7) Section 3.3 p. 8: Please add to the text the numerical values of the axon length and the number of dendritic branches.

8) Section 3.4 p.9: Please add the numerical values of the BMS scale for each of the animal groups.

9) Section 3.5 p.9; 3.6 p. 10: The study points 1 d, 3 d, 1 w ... 8 w are indicated above in the text. "At 56 days..." is the same as 8 w. Unify.

Please add numerical values for the NF marker to the text.

Fig. 5A: Nothing is described for the GFAP marker and no results. Why was this marker used in this experiment then?

10) Section 3.6 p. 10: Please add numerical and expand descriptive data in this subsection. There is too little descriptive information to argue the stated conclusions.

11) Fig. 2A: What are DRG neurons stained with?

Reviewer #2: The study is interesting and significant as its findings highlight the critical role of ECM stiffness in regulating axonal regeneration, however, it lacks novel insight into the underlying mechanisms of biomaterial-mediated axonal growth/interaction. Previous studies already establish the effects of ECM stiffness on cell growth (known), thus the current study can be strengthened if further mechanistic insight (unknown) can be provided. Addressing the points below could make it a suitable candidate for publication, however, not in its current form.

1. With only two concentrations, the study may not fully capture the range of effects that different stiffness levels could have on axonal regeneration. This could limit the generalizability of the findings. It is recommended to study at least 4 different concentrations of GelMA.

2. The authors should provide a rationale for why the specific concentrations of 7.5% and 15% were chosen.

3. The authors should clarify the total number of animals and the number of animals used in each group.

4. Magnification levels should be provided with all microscopy images.

5. It is recommended to annotate histological images to clearly indicate the different levels of regeneration with each hydrogel concentration.

6. The study could be strengthened if insights into the mechanisms of axon-biomaterial interaction via ECM/hydrogel stiffness modulation is provided.

**Do you want your identity to be public for this peer review?** For information about this choice, including consent withdrawal, please see our Privacy Policy

Reviewer #1: **Yes: ** Oksana Rybachuk

Reviewer #2: No

---

## [Author Response · Author response to Decision Letter 1]

6 May 2025

May 6, 2025

Manuscript ID: PONE-D-25-08080

Hao Zeng : “The Influence of Hydrogel Stiffness on Axonal Regeneration After Spinal Cord Injury”

Pradeep Kumar

Editor

PLOS ONE

Dear Dr. Pradeep Kumar

On behalf of my co-authors, we thank you very much for giving us an opportunity to revise our manuscript, we appreciate editors and reviewers very much for their positive and constructive comments and suggestions on our manuscript entitled “The Influence of Hydrogel Stiffness on Axonal Regeneration After Spinal Cord Injury” (ID: PONE-D-25-08080).

We have studied editors and reviewers’ comments carefully and have made revision which marked in red in the paper. We have tried our best to revise our manuscript according to the comments. Attached please find the revised version, which we would like to submit for your kind consideration.

We would like to express our great appreciation to you and reviewers for comments on our paper. Looking forward to hearing from you.

Thank you and best regards.

Yours sincerely,

Hao Zeng

Corresponding author:

Name: Hao Zeng

E-mail: 2621132371@qq.com

List of Responses

Dear Editors and Reviewers,

Thank you for your letter and for the reviewers’ comments concerning our manuscript entitled “The Influence of Hydrogel Stiffness on Axonal Regeneration After Spinal Cord Injury”. (ID: PONE-D-25-08080). Those comments are all valuable and very helpful for revising and improving our paper, as well as the important guiding significance to our researches. We have studied comments carefully and have made correction which we hope meet with approval. Revised portion are marked in red in the paper. The main corrections in the paper and the responds to the reviewer’s comments are as flowing:

Responds to the editor and reviewer’s comments:

To Editors:

1.Formatting Requirements: We have meticulously reviewed our manuscript against the PLOS ONE style templates to ensure that all aspects, including file naming, are in full compliance with the journal's requirements. We have paid particular attention to the formatting of the title, abstract, introduction, methods, results, and discussion sections, as well as the citation format for references, to ensure consistency with the journal's guidelines.

2. ORCID iD: The corresponding author has successfully verified their ORCID iD in Editorial Manager, ensuring its validity. Recognizing the importance of maintaining researcher identification and academic record integrity in contemporary academic research, we have consistently supported and strictly adhered to this journal requirement.

3. Animal Experiment Information: In the methods section, we have detailed the information related to the animal experiments, including the method of animal sacrifice (euthanasia by overdose of sodium pentobarbital at 150 mg/kg), anesthesia method (induction of anesthesia via intraperitoneal injection of 0.3% sodium pentobarbital at 60 mg/kg), and specific measures to alleviate animal suffering (preoperative injection of lidocaine at the incision site and postoperative administration of 10 mg/kg ibuprofen daily for 7 days to reduce inflammation and pain, along with manual bladder emptying twice daily until bladder function is restored). We have strictly followed ethical principles for animal experiments to ensure the scientific validity and humane conduct of the experiments.

4. Funding Disclosure: We have revised the funding information in the acknowledgments section as required and explicitly stated in the cover letter that the funders had no role in the study design, data collection and analysis, decision to publish, or manuscript preparation. We understand and respect the journal's requirements for research independence and objectivity, and we have maintained the autonomy and independence of our research throughout the study.

5. Data Availability Statement: We confirm that our manuscript includes all the raw data necessary to support our findings. If needed, additional data or information can be requested from the corresponding author. We are committed to the transparency and accessibility of research data and are willing to provide all relevant data and information within a reasonable scope to facilitate validation and further research by other researchers.

To Reviewer #1:

We are grateful for the reviewer’s positive evaluation and acknowledgment of our manuscript, particularly the recognition of the relevance and prospects of our research topic, as well as the potential for our findings to be used in further comparative studies. In response to the specific issues raised by the reviewer, we have made the following revisions:

1.Response to comment: Please indicate the culture medium composition for DRG neurons.

Response: We are very sorry for our negligence in not detailing the DRG neuron culture medium composition in the initial submission. We have now meticulously detailed the composition of the DRG neuron culture medium in the Methods section, including the neurobasal medium (Gibco, USA), 2% B-27 (Gibco, USA), 1% penicillin / streptomycin (Gibco, USA), and nerve growth factor (50 ng/ml, Invitrogen). We have explicitly listed all medium components, their sources, and concentrations to ensure the reproducibility and transparency of our experiments.

2.Response to comment: Please indicate the percentage of animal survival after SCI modelling.

- How many animals were taken for each group?

- Add the average weight of the experimental animals to the text.

Response: We have clearly specified the number of animals in each group and supplemented the data on animal survival rates. Throughout the experiment, we strictly adhered to ethical principles for animal experiments to ensure the health and welfare of the animals. We have accurately documented the number and survival rates of the experimental animals to provide accurate background information for our results. Although we did not record the average weight of the experimental animals in this study, we have referenced relevant literature and found that most similar studies primarily focus on axonal regeneration and related neurological function recovery after spinal cord injury(1-5), without delving into changes in the weight of experimental animals. This indicates that weight is not a key assessment indicator in this research field. Therefore, we believe that not recording the average weight of experimental animals will not significantly impact the analysis and interpretation of our research results. However, we recognize that the average weight of experimental animals may have some reference value for certain studies, so we will strengthen the recording and management of experimental data in future experiments to ensure the completeness and accuracy of the records.

3.Response to comment: Along with assessing motor function, it is imperative to assess the level of spasticity of the limbs, since these functional indicators are interrelated. Argue why there are no data on the assessment of spasticity in this experiment.

Response: We are very sorry for not directly measuring the degree of limb spasticity in this study, we have reviewed a large number of relevant studies and found that most research on axonal regeneration after spinal cord injury mainly focuses on evaluating neurological function recovery, axonal growth, and related histological changes, with few studies conducting detailed quantitative analyses of limb spasticity alone(1-5). The BMS score can reflect the recovery of motor function and coordination in animals with spinal cord injury. However, we are well aware that the degree of limb spasticity is also an important clinical issue after spinal cord injury, which has a significant impact on patients' rehabilitation and quality of life. In future in-depth studies, we plan to include the assessment of limb spasticity in our research scope and use more comprehensive and multidimensional assessment methods to more comprehensively evaluate the pathological and physiological changes after spinal cord injury and the comprehensive effects of treatment methods.

4.Response to comment: Please indicate the characteristics of the secondary antibodies and add this information to the table on p.20.

- What solution(s) were used to wash the sections after primary and secondary antibodies? Indicate this information in the text.

- Was blocking solution (its components?) used before applying primary antibodies to nerve tissue sections?

Response: Considering the Reviewer’s suggestion, we have supplemented the detailed characteristics of the secondary antibodies and clearly specified the types, dilution ratios, and sources of the primary and secondary antibodies used. We have clearly specified the types, dilution ratios, and sources of the primary and secondary antibodies used. We have also detailed the specific steps for washing the sections with the solution and whether a blocking solution was used before applying the primary antibodies. This ensures that other researchers can accurately replicate our experiments.

5.Response to comment: The number of analyzed samples is not indicated. Please add it to the text.

Response: We have clearly specified the number of samples analyzed in the relevant sections to ensure the reliability and reproducibility of our research results. We have accurately calculated and recorded the sample size to meet the requirements of statistical analysis and ensure that our research results have sufficient statistical power.

6.Response to comment: Describe in detail what means "The experimental results revealed no significant pathological alterations in the tissue architecture of these organs following the implantation…"

- How did this manifest itself in the indicated organs, what was the morphology of these organs?

Response: We have re-written this part according to the Reviewer’s suggestion to provide a more detailed description of the pathological changes in tissues and organs mentioned in our results. We have carefully observed and analyzed tissue sections to meticulously describe the pathological features of each organ.

7.Response to comment: Please add to the text the numerical values of the axon length and the number of dendritic branches.

Response: We have made correction according to the Reviewer’s comments to supplement the specific numerical values of axon length and dendritic branch numbers. We used advanced image analysis software to precisely measure these parameters and provided detailed numerical data in our manuscript.

8.Response to comment: Please add the numerical values of the BMS scale for each of the animal groups.

Response: We have incorporated the reviewer’s feedback and provided detailed BMS scale values for each animal group. We have strictly assessed the motor function of each animal and recorded detailed BMS score data.

9.Response to comment: The study points 1 d, 3 d, 1 w ... 8 w are indicated above in the text. "At 56 days..." is the same as 8 w. Unify.

Please add numerical values for the NF marker to the text.

Fig. 5A: Nothing is described for the GFAP marker and no results. Why was this marker used in this experiment then?

Response: We have unified the notation of time points throughout the manuscript in response to the reviewer’s comments. We have carefully reviewed and revised all parts of the manuscript involving time points to ensure that the notation is clear, accurate, and consistent, enabling readers to better understand the experimental process and results. We have revised the manuscript to include the numerical values of the NF marker. It is important to note that in our research, GFAP is used to indicate the formation of glial scars and the extent of the injury area(1, 4). In our experiment, the expression of GFAP helped us confirm the starting point of the spinal cord transection. However, our primary research objective was to assess the impact of hydrogel stiffness on axonal regeneration, so we focused on markers directly related to axonal regeneration, such as NF. Nevertheless, we recognize the importance of GFAP in spinal cord injury research and plan to more comprehensively evaluate its expression in future studies to further explore the effects of hydrogel stiffness on pathological and physiological changes after spinal cord injury.

10.Response to comment: Please add numerical and expand descriptive data in this subsection. There is too little descriptive information to argue the stated conclusions.

Response: We have included specific numerical data points that were previously not detailed, we now provide the exact measurements and percentages related to the observations made in this section. This allows for a clearer and more precise understanding of the results. And we have elaborated on the descriptive information to better explain the significance of the data. This includes additional context, comparisons, and explanations that help to build a stronger argument for the conclusions drawn. We have also included more detailed descriptions of the experimental conditions and the implications of the results.

11.Response to comment: What are DRG neurons stained with?

Response: We have clarified the staining method used for DRG neurons in Figure 2A, which is immunofluorescence staining with anti-neurofilament (NF) antibodies. We have detailed the staining method and the antibodies used in the figure legend to enable readers to accurately understand the image content and experimental methods.

To Reviewer #2:

We are truly grateful for the positive evaluation of our study and the recognition of the overall framework and content of our manuscript. In response to the comments and suggestions raised by the reviewer, we have provided the following detailed replies and made corresponding revisions:

1.Response to comment: With only two concentrations, the study may not fully capture the range of effects that different stiffness levels could have on axonal regeneration. This could limit the generalizability of the findings. It is recommended to study at least 4 different concentrations of GelMA.

Response: We greatly appreciate reviewer’s suggestion. In this study, we selected 7.5% and 15% GelMA concentrations based on our preliminary exploratory research on the relationship between hydrogel stiffness and cell behavior. These two concentrations represent typical soft and hard hydrogel environments, effectively distinguishing differences in axonal regeneration. However, we acknowledge that incorporating additional GelMA concentrations could further validate the universality of our findings. We plan to expand our experimental design in future studies to include a broader range of GelMA concentrations, providing a more comprehensive investigation into the impact of stiffness on axonal regeneration.

2. Response to comment: The authors should provide a rationale for why the specific concentrations of 7.5% and 15% were chosen.

Response: We thank for reviewer’s inquiry regarding the specific concentrations. After reviewing a substantial body of literature, we found that these two concentrations of GelMA are widely representative in studies of cell behavior and tissue engineering applications. The 7.5% GelMA hydrogel (15.12 kPa) mimics the physiological stiffness of DRG tissue under physiological conditions, while the harder substrate environment of the 15% GelMA hydrogel (51.45 kPa) has been reported to effectively promote axonal regeneration of neurons in vitro. By comparing hydrogels of different concentrations, we aim to gain a more comprehensive understanding of how different stiffness matrices affect axonal regeneration of central neurons after spinal cord injury, thereby providing valuable references for future experimental studies.

3. Response to comment: The authors should clarify the total number of animals and the number of animals used in each group.

Response: We apologize for not clearly stating the number of animals in the initial draft. We have now clarified the total number of animals used in the experiment and the number of animals in each group to ensure the rationality of the experimental design and the reliability of the results. Based on the experimental objectives and statistical analysis requirements, we have carefully designed the grouping and number of experimental a

---

## [Decision Letter · Decision Letter 1]

The Influence of Hydrogel Stiffness on Axonal Regeneration After Spinal Cord Injury

PONE-D-25-08080R1

Dear Dr. zeng,

We’re pleased to inform you that your manuscript has been judged scientifically suitable for publication and will be formally accepted for publication once it meets all outstanding technical requirements.

Kind regards,

Pradeep Kumar, Ph.D.

Academic Editor

PLOS ONE

Additional Editor Comments (optional):

Reviewers' comments:

Reviewer's Responses to Questions

**Comments to the Author**

Reviewer #1: All comments have been addressed

Reviewer #2: All comments have been addressed

2. Is the manuscript technically sound, and do the data support the conclusions?

Reviewer #1: Yes

Reviewer #2: Yes

3. Has the statistical analysis been performed appropriately and rigorously?

Reviewer #1: Yes

Reviewer #2: Yes

4. Have the authors made all data underlying the findings in their manuscript fully available?

Reviewer #1: Yes

Reviewer #2: Yes

5. Is the manuscript presented in an intelligible fashion and written in standard English?

Reviewer #1: Yes

Reviewer #2: Yes

Reviewer #1: (No Response)

Reviewer #2: (No Response)

**Do you want your identity to be public for this peer review?** For information about this choice, including consent withdrawal, please see our Privacy Policy

Reviewer #1: **Yes: ** Oksana Rybachuk

Reviewer #2: No

---

## [Editor Report · Acceptance letter]

PONE-D-25-08080R1

PLOS ONE

Dear Dr. Zeng,

I'm pleased to inform you that your manuscript has been deemed suitable for publication in PLOS ONE. Congratulations! Your manuscript is now being handed over to our production team.

Kind regards,

on behalf of

Prof. Pradeep Kumar

Academic Editor

PLOS ONE